# Risk of heart disease following treatment for breast cancer – results from a population-based cohort study

**Haomin Yang[1,2]\*, Nirmala Bhoo-Pathy[3], Judith S Brand[4], Elham Hedayati[5], Felix Grassmann[2,6], Erwei Zeng[2], Jonas Bergh[5,7,8], Weiwei Bian[2], Jonas F Ludvigsson[2,9], Per Hall[2,10], Kamila Czene[2]**

[1]Department of Epidemiology and Health Statistics, School of Public Health, Fujian Medical University, Fuzhou, China; [2]Department of Medical Epidemiology and Biostatistics, Karolinska Institutet, Stockholm, Sweden; [3]Centre for Epidemiology and Evidence-Based Practice, Faculty of Medicine, University of Malaya, Kuala Lumpur, Malaysia; [4]Clinical Epidemiology and Biostatistics, School of Medical Sciences, Örebro University, Örebro, Sweden; [5]Department of Oncology-Pathology, Karolinska Institutet, Stockholm, Sweden; [6]Health and Medical University, Potsdam, Germany; [7]Breast Cancer Center, Karolinska University Hospital, Stockholm, Sweden; [8]Karolinska Comprehensive Cancer Center, Stockholm, Sweden; [9]Department of Pediatrics, Örebro University Hospital, Örebro, Sweden; [10]Department of Oncology, Södersjukhuset, Stockholm, Sweden

**\*For correspondence:**
haomin.yang@ki.se

## Abstract

**Background:** There is a rising concern about treatment-associated cardiotoxicities in breast cancer patients. This study aimed to determine the time- and treatment-specific incidence of arrhythmia, heart failure, and ischemic heart disease in women diagnosed with breast cancer.

**Methods:** A register-based matched cohort study was conducted including 8015 breast cancer patients diagnosed from 2001 to 2008 in the Stockholm-Gotland region and followed up until 2017. Time-dependent risks of arrhythmia, heart failure, and ischemic heart disease in breast cancer patients were assessed using flexible parametric models as compared to matched controls from general population. Treatment-specific effects were estimated in breast cancer patients using Cox model.

**Results:** Time-dependent analyses revealed long-term increased risks of arrhythmia and heart failure following breast cancer diagnosis. Hazard ratios (HRs) within the first year of diagnosis were 2.14 (95% CI = 1.63–2.81) for arrhythmia and 2.71 (95% CI = 1.70–4.33) for heart failure. HR more than 10 years following diagnosis was 1.42 (95% CI = 1.21–1.67) for arrhythmia and 1.28 (95% CI = 1.03–1.59) for heart failure. The risk for ischemic heart disease was significantly increased only during the first year after diagnosis (HR = 1.45, 95% CI = 1.03–2.04). Trastuzumab and anthracyclines were associated with increased risk of heart failure. Aromatase inhibitors, but not tamoxifen, were associated with risk of ischemic heart disease. No increased risk of heart disease was identified following locoregional radiotherapy.

**Conclusions:** Administration of systemic adjuvant therapies appears to be associated with increased risks of heart disease. The risk estimates observed in this study may aid adjuvant therapy decision-making and patient counseling in oncology practices.

**Funding:** This work was supported by the Swedish Research Council (grant no: 2018-02547); Swedish Cancer Society (grant no: CAN-19-0266); and FORTE (grant no: 2016-00081).

## Editor's evaluation

We feel that your work will be of interest to breast medical oncologists, cardiologists, and primary care providers who treat patients with breast cancer. We commend you for this study, which achieves its goal of identifying the incidence and hazard ratio of cardio-toxicity associated with breast cancer treatment within a general breast cancer population. The international nature of your collaborative study along with its large patient cohort size and long horizontal follow up are quite attractive features in solidifying previous findings and discovering future areas of exploration.

## Introduction

The use of adjuvant systemic therapies at least halves the risk of dying from breast cancer (*Early Breast Cancer Trialists' Collaborative Group (EBCTCG), 2005*; *Dowsett et al., 2015*; *Goldvaser et al., 2019*; *Gray et al., 2019*). Nowadays, 80% of the breast cancer patients survive for at least 10 years and many will become long-term survivors. There are, however, concerns about therapy-associated late adverse health effects, including cardiovascular events (*Khouri et al., 2012*). Use of common (neo-)adjuvant therapies for breast cancer has been associated with an increased risk of heart diseases including heart failure, arrhythmias, and ischemic heart disease (*Darby et al., 2013*; *Doyle et al., 2005*; *Harris et al., 2006*; *Hooning et al., 2007*; *Pinder et al., 2007*; *Taylor et al., 2017*; *Yeh and Bickford, 2009*). However, this evidence mostly comes from studies focusing on specific subgroups of patients based on age, cancer stage, or treatment regimen.

Although the benefits of radiotherapy far outweigh the risk of heart diseases, some studies have found increased incidence of and mortality due to heart disease in women subjected to radiotherapy techniques (*Darby et al., 2013*; *Taylor et al., 2017*). Knowledge on cardiotoxic effects of anthracycline-based chemotherapy regimens have led to lowering of doses and less use of bolus injections to reduce peak concentrations of anthracyclines (*Foukakis et al., 2016*), but it is estimated that the risk of heart failure associated with anthracyclines remains increased for standard low-dose group as compared to non-users (*Chung et al., 2020*). While trastuzumab has been shown to reduce the risk of breast cancer mortality at 11 years of follow-up (*Cameron et al., 2017*), evidence on its cardiotoxicity is conflicting (*Bowles et al., 2012*; *Papakonstantinou et al., 2020*). In addition, recent evidence suggests that use of aromatase inhibitors in women with hormone receptor positive breast cancer may increase the risk of heart failure, compared to tamoxifen (*Khosrow-Khavar et al., 2020*).

Risk assessment of immediate and later occurring heart disease events following breast cancer is important for the planning of cardiac surveillance programs and possible prophylactic pharmacotherapy. Here, we report the risks of heart diseases in a cohort representative of the general breast cancer population with long-term follow-up. We specifically aimed to assess risks of heart diseases by time since diagnosis, and according to adjuvant treatments.

## Materials and methods

### Breast cancer cohort

This study took advantage of the Stockholm-Gotland Breast Cancer Register comprising all women diagnosed with primary invasive breast cancer between 2001 and 2008 in the Stockholm-Gotland region. The Stockholm-Gotland Breast Cancer Register has about 99% completeness and provides detailed information on tumor and treatment characteristics, as well as routine follow-up on locoregional recurrences and distant metastases (*Colzani et al., 2011*; *Wigertz et al., 2012*). Detailed description of the breast cancer cohort can be found elsewhere (*Holm et al., 2016*). We included all patients diagnosed with non-metastatic breast cancer (stages I–III) and without prior diagnosis of heart disease at age 25–75 years (N = 8015). To compare the risk of heart diseases after breast cancer diagnosis, we randomly sampled up to 10 women from the general female population in Stockholm-Gotland region matched on year of birth (*Appendix 1—figure 1*). Each reference individual was alive and free of breast cancer on the date of the matched patient's diagnosis of breast cancer (the index date).

The matched cohort was linked through the unique personal identity number to the Swedish Cancer Register, Patient Register, Cause of Death Register and Migration Register, and follow-up started from

the index date until the date of heart disease diagnosis, emigration, death, breast cancer relapse, or end of follow-up (December 31, 2017), whichever occurred first. We also performed linkage with the Prescribed Drug Register which contains data on all drugs dispensed from Swedish pharmacies from July 2005 onward to validate the use of hormone therapy.

The study was approved by the Regional Ethical Review Board in Stockholm.

### Heart diseases

We identified the following heart diseases according to relevant ICD (International Classification of Disease) codes in the Swedish Patient Register and the Cause of Death Register (*Appendix 2—table 1*): heart failure (ICD-10: I50, ICD-9: 428A, 428B, 428X), arrhythmias (ICD-10: I47–I49, ICD-9: 427), and ischemic heart disease (ICD-10: I20–I25, ICD-9: 410–414). We included both inpatient and outpatient diagnoses, as well as cause of death records in our outcome definition, except for myocardial infarction, which was based on inpatient and cause of death records solely. To ensure specificity of the outcomes studied, only primary, and not underlying, diagnoses were considered for analyses.

### Breast cancer treatment specifics

We extracted the treatment data from the Stockholm-Gotland Breast Cancer Register. As ~90% of HER-2 positive cancers were treated with trastuzumab between 2005 and 2008 in the Stockholm-Gotland region and the Swedish Prescribed Drug Register does not cover data on treatment with trastuzumab, HER-2 positivity was used as a proxy when no registry data on trastuzumab was available during this time period (30% of the HER-2 positive patients had missing information on trastuzumab). Data on adjuvant endocrine therapy was verified against the Prescribed Drug Register and categorized into tamoxifen and/or aromatase inhibitor use. Since radiotherapy to the left breast has, in particular, been implicated in heart complications, radiotherapy was categorized according to tumor laterality (left vs. right). Bilateral tumors were coded separately in this analysis. Chemotherapy was coded as anthracycline-based, anthracycline plus taxane-based, cyclophosphamide, methotrexate, and fluorouracil (CMF)-based regimens.

### Covariates

The Stockholm Breast Cancer Register contains data on date of diagnosis, menopausal status at diagnosis, and type of surgery (breast conserving surgery vs. mastectomy). Tumor characteristics were also retrieved from this register, including tumor size (T), regional lymph node involvement (N), and presence of metastases (M), all from pathology records and summarized in TNM stage as defined according to the *American Joint Committee on Cancer, 2010*. Information on inpatient comorbidities at diagnosis was extracted through the Swedish Patient Register and summarized into the Charlson comorbidity index (CCI) score, a widely used method for classifying chronic comorbid conditions (*Charlson et al., 1987*). To account for the potential confounding effect from tobacco abuse, chronic pulmonary disease, and hypertension in the associations, we further identified associated diagnoses before cancer using ICD codes from the patient register (*Appendix 2—table 1*).

### Statistical analyses

We compared the risk of heart diseases in breast cancer patients with that observed in the matched cohort, using flexible parametric model (FPM) with time since index date as underlying time scale. The FPM is similar to the Cox proportional hazards model in that it provides a hazard ratio (HR) as measure of association. In our analysis, a restricted cubic spline with four internal and two boundary knots (five degrees of freedom) placed at quintiles of the event times was used in the FPM for the baseline hazard. The key advantage of FPM is that non-proportional hazards can easily be fitted by adding a second spline for the interaction with time. Considering the correlation within the matched clusters, a shared frailty term (as random effects) was incorporated into the model and the maximum (penalized) marginal likelihood method was used to estimate the regression coefficients and the variance for the frailty. Aalen-Johansen estimation was used to assess the cumulative incidences of heart diseases in breast cancer patients and matched reference individuals, while other causes of death were considered as competing events.

Next, we studied the association of adjuvant breast cancer therapy with heart disease risk in breast cancer patients using Cox proportional hazards models. We adjusted these analyses for age

and year of diagnosis (model 1), and additionally for menopausal status at diagnosis, cancer stage, type of surgery, CCI score, hypertension, chronic pulmonary disease, and tobacco abuse. All treatment-specific models were mutually adjusted for adjuvant therapies. Considering the possible selection bias in the administration of radiotherapy (*Wadsten et al., 2018*), the analysis for radiotherapy only included patients receiving radiotherapy, making a comparison between left-sided, right-sided, and both-sided breast cancer. Multiple imputation with chained equations was used to deal with the treatment categories with missing information. We replaced the missing data with 10 rounds of imputations and all the covariates were included in the imputation model. Considering the time-dependent effect of treatment, we separated the analysis according to different follow-up periods, within 10 years after breast cancer diagnosis and beyond, respectively.

All statistical analyses were performed using STATA version 15.1.

## Results

Descriptive characteristics of the study population are described in *Table 1*. The median age at breast cancer diagnosis was 59 years, with 74.3% of patients aged less than 65 years (*Table 1*). Approximately 40% of all patients received adjuvant chemotherapy, with anthracyline-based regimens being most frequently administered. Endocrine therapy was received by more than 80% of patients (constituting mainly of tamoxifen). Of all patients diagnosed between mid-2005 and 2008, 13% were coded as receivers of trastuzumab.

Over a median follow-up of 10.8 years (interquartile range = 6.5 years), arrhythmias were the most frequently reported heart disease (n = 570), followed by ischemic heart disease (n = 307) and heart failure (n = 243). The cumulative incidence of arrhythmias, ischemic heart disease, and heart failure in the breast cancer patients cohort was 11.0%, 5.7%, and 4.8% after 15 years of follow-up, while the cumulative incidence was 8.2%, 5.9%, and 3.8% in the matched cohort, respectively (*Appendix 2—table 2* and *Appendix 3—figure 1*).

*Figure 1* shows the time-dependent risks of heart diseases in breast cancer patients compared to the reference population. A short-term increase in risks of arrhythmia and heart failure was found in breast cancer patients (*Table 2*, *Figure 1*, HR at first year for arrhythmia = 2.14; 95% CI = 1.63–2.81, for heart failure = 2.71; 95% CI = 1.70–4.33,

**Table 1.** Descriptive characteristics of the study population.

| Characteristics | Overall (N = 8015) |
|---|---|
| Year of diagnosis % (N) | |
| 2001–2002 | 25.6 (2072) |
| 2003–2004 | 24.4 (1952) |
| 2005–2006 | 25.8 (2064) |
| 2007–2008 | 24.0 (1927) |
| Age at diagnosis % (N) | |
| <50 years | 23.0 (1842) |
| 50–65 years | 51.3 (4111) |
| >65 years | 25.7 (2062) |
| Menopausal status % (N) | |
| Premenopausal | 30.4 (2289) |
| Postmenopausal | 69.6 (5251) |
| *Missing (N)* | *475* |
| Stage % (N) | |
| Stage I | 48.2 (3690) |
| Stage II | 39.4 (3014) |
| Stage III | 12.4 (948) |
| *Missing (N)* | *363* |
| Tumor size % (N) | |
| ≤2 cm | 25.7 (2000) |
| 2–5 cm | 44.0 (3418) |
| > 5 cm | 30.3 (2351) |
| *Missing (N)* | *246* |
| Comorbidity % (N) | |
| None | 90.1 (7222) |
| 1 | 5.5 (439) |
| ≥2 | 4.4 (354) |
| History of hypertension | |
| No | 94.6 (7579) |
| Yes | 5.4 (436) |
| History of chronic pulmonary disease or tobacco abuse | |
| No | 96.6 (7746) |
| Yes | 3.4 (269) |
| Surgery % (N) | |
| No | 1.0 (79) |
| Yes, breast-conserving | 60.7 (4852) |
| Yes, mastectomy | 38.3 (3058) |

*Table 1 continued on next page*

*Table 1 continued*

| Characteristics | Overall (N = 8015) |
|---|---|
| *Missing (N)* | *26* |
| Radiotherapy % (N) | |
| No | 22.5 (1774) |
| Yes, left-sided | 37.5 (2962) |
| Yes, right-sided | 39.1 (3088) |
| Yes, both-sided | 0.85 (67) |
| *Missing (N)* | *124* |
| Chemotherapy % (N) | |
| No | 58.5 (4604) |
| Yes | 41.5 (3262) |
| *Missing (N)* | *149* |
| Hormone therapy % (N) | |
| No | 18.10 (1424) |
| Yes, tamoxifen | 53.0 (4247) |
| Yes, aromatase inhibitors | 19.3 (1550) |
| Yes, type unknown | 8.1 (645) |
| *Missing (N)* | *149* |
| Trastuzumab therapy % (N) * | |
| No | 87.3 (2180) |
| Yes | 12.7 (316) |
| *Missing (N)* | *1497* |

*Missingness on individual variables is less than 5%, except for menopausal status (5.9%, N = 488). Treatment-specific analysis of trastuzumab was restricted to patients diagnosed between 2005 and 2008 (missingness = 37.5%, N = 1497).

respectively). An elevated risk of ischemic heart disease was also observed in the first year after diagnosis (HR = 1.45; 95% CI = 1.03–2.04), followed by a decline with increasing follow-up time. Interestingly, increased risks for arrhythmia and heart failure were noted beyond 10 years (*Table 2*, *Figure 1*, HR for arrhythmia = 1.42; 95% CI = 1.21–1.67; HR for heart failure = 1.28; 95% CI = 1.03–1.59).

*Table 3* presents the HRs in breast cancer patients by adjuvant therapy. Patients treated with chemotherapy and trastuzumab had an increased risk of heart failure compared to patients not receiving these treatments (for patients receiving anthracyclines-based chemotherapy, HR = 1.74, 95% CI = 1.20–2.52; for patients receiving trastuzumab, HR = 2.34, 95% CI = 1.05–5.22) (*Table 3*). The risk of heart failure in patients receiving anthracyclines potentially persisted for 10 years beyond the diagnosis of breast cancer (HR = 1.66, 95% CI = 0.86–3.19, *Appendix 2—table 3*). Receipt of aromatase inhibitors was associated with risk of ischemic heart disease (HR = 1.52, 95% CI = 1.03–2.26 compared to patients without hormonal therapy), but not with risk of heart failure or arrhythmias (*Table 3*). A direct comparison between left-sided and right-sided radiotherapy showed no evidence of an association of radiotherapy with heart disease except for a somewhat increased risk of ischemic heart disease in women with left-sided breast cancer, particularly after 10 years after cancer diagnosis, although not statistically significant (*Appendix 2—table 3*, HR = 1.16, 95% CI = 0.89–1.51; HR for the risk beyond 10 years = 1.29, 95% CI = 0.70–2.37).

## Discussion

In a population-based setting, we demonstrated that the incidence of heart disease in breast cancer patients was significantly higher than the incidence observed in matched reference individuals from the general population. The risks of arrhythmia and heart failure were increased even beyond 10 years after diagnosis. Receipt of trastuzumab, as well as administration of anthracycline±taxane-based regimens, was independently associated with an increased risk of heart failure, while receipt of aromatase inhibitor therapy was associated with an increased risk of ischemic heart disease.

We found an increased risk of arrhythmia and heart failure in breast cancer patients as compared with the matched reference individuals from the general population, which is similar to the risk of heart failure reported by a previous Dutch study (*Hooning et al., 2007*), indicating the generalizability of our findings to European countries. However, as patients in our cohort were aged between 25 and 75 years, caution is needed when generalizing these findings to older patients, who may have more comorbidities.

Analysis by time since diagnosis revealed long-term increased risks of arrhythmia and heart failure following breast cancer diagnosis, suggesting that a longitudinal cardiac monitoring schedule might be helpful to improve cardiac health in breast cancer patients. As the long-term risk was observed for heart failure but not ischemic heart disease, the cardiotoxic effect of chemotherapy might be mainly on the myocardium mediated by the effect of DNA double-strand breaks through topoisomerase

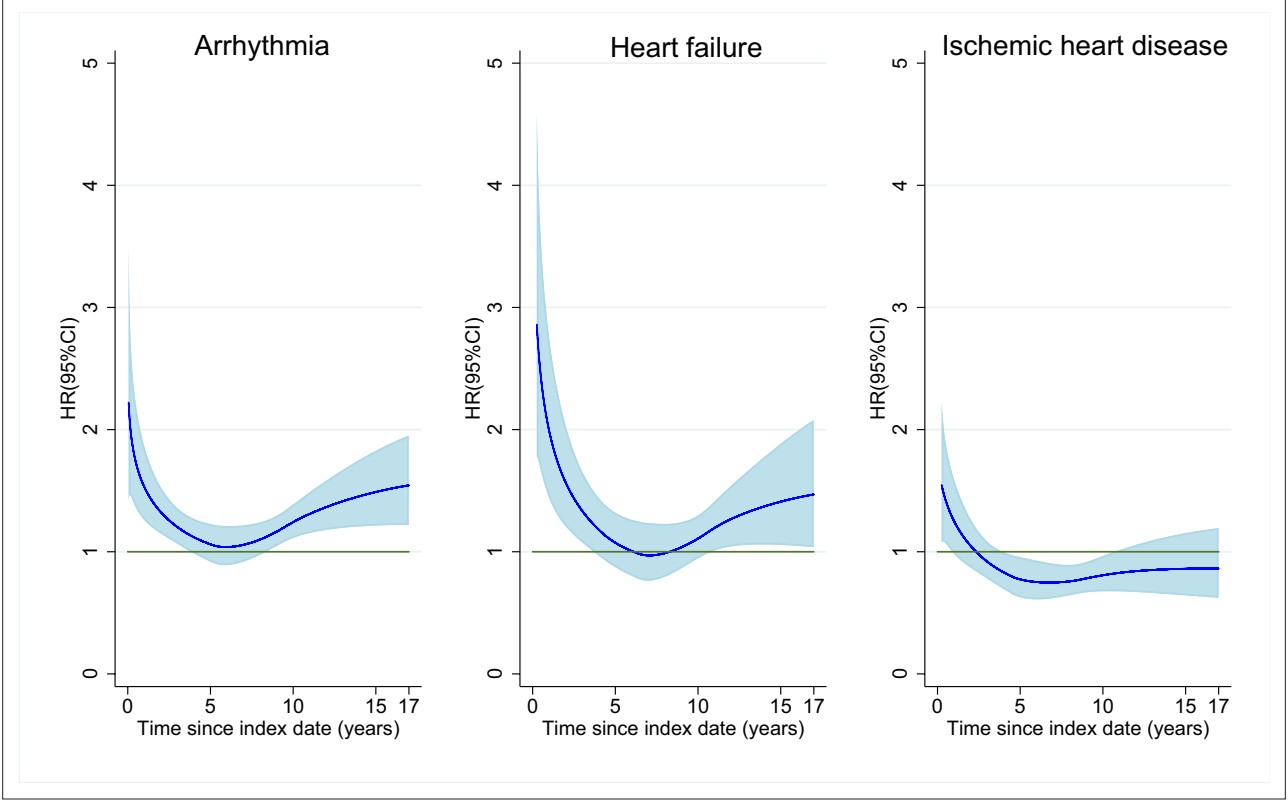

**Figure 1.** Time-dependent hazard ratio of heart diseases in breast cancer patients compared to age-matched women. In all models, time since index date was the underlying time scale and a restricted cubic spline with four internal and two boundary knots (five degrees of freedom) placed at quintiles of the event times was used for the baseline hazard. Time-dependent effects were modelled by adding interaction terms with time using a second spline with three degrees of freedom.

The online version of this article includes the following source data for figure 1:

**Source data 1.** Summary data for *Figure 1*.

**Table 2.** Hazard ratios for heart diseases in breast cancer patients compared to the matched cohort.

| | | Arrhythmia | | Heart failure | | Ischemic heart disease |
|---|---|---|---|---|---|---|
| | No. | HR (95% CI) | No. | HR (95% CI) | No. | HR (95% CI) |
| Time since diagnosis | | | | | | |
| <1 year | 64 | **2.14 (1.63–2.81)** | 22 | **2.71 (1.70–4.33)** | 38 | **1.45 (1.03–2.04)** |
| 1–2 years | 34 | 1.08 (0.76–1.53) | 19 | **2.07 (1.27–3.37)** | 34 | 1.12 (0.79–1.61) |
| 2–5 years | 107 | 1.07 (0.88–1.30) | 38 | 1.14 (0.82–1.59) | 72 | 0.84 (0.66–1.07) |
| 5–10 years | 204 | 1.13 (0.98–1.30) | 78 | 1.02 (0.81–1.29) | 104 | 0.82 (0.67–1.00) |
| 10–17 years | 161 | **1.42 (1.21–1.67)** | 86 | **1.28 (1.03–1.59)** | 59 | 0.79 (0.61–1.03) |

Abbreviations: No. = number of cases. HR = hazard ratio. CI = confidence interval. The HRs are estimated using flexible parametric model and conditioned on matching criteria (year of birth). In all models, time since index date was the underlying time scale and a restricted cubic spline with four internal and two boundary knots (five degrees of freedom) placed at quintiles of the event times was used for the baseline hazard. All analyses were stratified by time since index date. Statistically significant results with p-value<0.05 are bolded.

**Table 3.** Hazard ratios for heart diseases in breast cancer patients by different adjuvant therapies.

| Treatment variables | Total No. | HR (95% CI) for arrhythmia | | | HR (95% CI) for heart failure | | | HR (95% CI) for ischemic heart disease | | |
|---|---|---|---|---|---|---|---|---|---|---|
| | | N | Model 1 | Model 2 | N | Model 1 | Model 2 | N | Model 1 | Model 2 |
| **Radiotherapy** | | | | | | | | | | |
| Right-sided | 2948 | 216 | REF (1.00) | REF (1.00) | 80 | REF (1.00) | REF (1.00) | 99 | REF (1.00) | REF (1.00) |
| Left-sided | 3075 | 213 | 0.93 (0.77–1.13) | 0.93 (0.77–1.13) | 87 | 1.03 (0.76–1.40) | 1.07 (0.79–1.45) | 119 | 1.14 (0.87–1.49) | 1.16 (0.89–1.51) |
| Both-sided | 67 | 4 | 0.65 (0.24–1.76) | 0.63 (0.23–1.70) | 4 | 1.81 (0.66–4.95) | 1.75 (0.63–4.85) | 3 | 1.08 (0.34–3.40) | 0.98 (0.31–3.11) |
| **Chemotherapy** | | | | | | | | | | |
| No. | 4604 | 376 | REF (1.00) | REF (1.00) | 133 | REF (1.00) | REF (1.00) | 207 | REF (1.00) | REF (1.00) |
| Anthracyclines-based | 1426 | 83 | 1.18 (0.97–1.45) | 1.08 (0.84–1.39) | 51 | 2.30 (1.70–3.10) | 1.74 (1.20–2.52) | 44 | 1.31 (0.99–1.72) | 1.29 (0.92–1.81) |
| Anthracyclines + taxanes | 286 | 10 | 1.15 (0.63–2.10) | 1.01 (0.53–1.92) | 13 | 4.71 (2.57–8.63) | 3.09 (1.55–6.14) | 6 | 1.36 (0.63–2.93) | 1.29 (0.57–2.95) |
| CMF | 96 | 6 | 1.08 (0.55–2.14) | 1.04 (0.51–2.12) | 3 | 1.21 (0.36–4.00) | 0.90 (0.27–3.04) | 2 | 1.11 (0.49–2.51) | 1.04 (0.46–2.36) |
| **Hormone therapy** | | | | | | | | | | |
| No. | 1424 | 89 | REF (1.00) | REF (1.00) | 49 | REF (1.00) | REF (1.00) | 43 | REF (1.00) | REF (1.00) |
| Tamoxifen | 4247 | 298 | 0.91 (0.72–1.15) | 0.95 (0.74–1.23) | 99 | 0.57 (0.41–0.79) | 0.84 (0.58–1.20) | 142 | 0.91 (0.65–1.27) | 1.03 (0.71–1.48) |
| Aromatase inhibitors | 1550 | 119 | 1.02 (0.77–1.35) | 0.99 (0.74–1.32) | 61 | 1.05 (0.71–1.56) | 1.08 (0.72–1.61) | 85 | 1.53 (1.05–2.24) | 1.52 (1.03–2.26) |
| **Trastuzumab*** | | | | | | | | | | |
| No. | 2137 | 126 | REF (1.00) | REF (1.00) | 38 | REF (1.00) | REF (1.00) | 60 | REF (1.00) | REF (1.00) |
| Yes | 304 | 17 | 1.15 (0.70–1.92) | 1.50 (0.79–2.83) | 13 | 3.05 (1.62–5.76) | 2.34 (1.05–5.22) | 12 | 1.75 (0.94–3.27) | 1.83 (0.85–3.96) |

Total No. refers to the total number of patients. N events refers to the number of observed cases. HR = hazard ratio; CI = confidence interval; CMF = cyclophosphamide, methotrexate, and fluorouracil; REF: reference women in the matched controls from the general population. Hazard ratios are estimated from Cox proportional hazards models with time since diagnosis as underlying time scale. Hazard ratios for model 1 are adjusted for age and calendar period. Hazard ratios for model 2 are multivariable adjusted including age at diagnosis, year of diagnosis, menopausal status, Charlson comorbidity index, clinical stage, tumor size, type of surgery, history of hypertension, chronic pulmonary disease and tobacco abuse, and all treatment variables listed in the table.

*Treatment-specific analysis of trastuzumab was restricted to patients diagnosed between 2005 and 2008.

(Top) 2β, but not the cardiac vessels (*Lyu et al., 2007*). The finding that risk of ischemic heart disease in breast cancer patients was only transiently elevated after diagnosis is not unexpected, considering the emotional distress of dealing with a new cancer diagnosis in the patients, which may lead to higher short-term rates of ischemic heart disease (*Fang et al., 2012*; *Schoormans et al., 2016*). In addition, surgery after breast cancer diagnosis might increase the risk of arterial thromboembolism (*Gervaso et al., 2021*), which includes myocardial infarction, and the effect appears to attenuate 1 year after diagnosis (*Navi et al., 2017*; *Navi et al., 2019*). The long-term lower risk of ischemic heart disease in breast cancer patients compared to age-matched women might be explained by the opposite role of reproductive factors in breast cancer and ischemic heart disease. Women with younger age at menarche and older age at menopause were associated with increased risk of breast cancer, while decreased risk of ischemic heart disease were found among these women (Collaborative Group on Hormonal Factors in *Breast, 2012*; *Okoth et al., 2020*).

The findings that administration of anthracycline±taxanes, as well as trastuzumab, is associated with increased risks of heart failure are in agreement with previous observational studies (*Bowles et al., 2012*; *Chavez-MacGregor et al., 2013*; *Doyle et al., 2005*; *Du et al., 2009*; *Goldhar et al., 2016*; *Thavendiranathan et al., 2016*) and clinical trials (*Chen et al., 2011*) in specific subgroups of breast cancer patients. While (poly)chemotherapy is thought to lead to heart failure through a plethora of mechanisms that damage cardiomyocytes (*Yeh and Bickford, 2009*), taxanes may also interrupt the metabolism and excretion of anthracyclines, hence accentuating their toxicity (*Bird and Swain, 2008*). The cardiotoxic effect of trastuzumab meanwhile may be explained by inhibition of the HER-2 receptors in myocytes, which activates the mitochondrial apoptosis pathway through modulation of Bcl-xL and -xS, which regulates cell development and growth (*Grazette et al., 2004*; *Yeh and Bickford, 2009*). As there is no cardiac monitoring for chemotherapy in routine clinical practice and cardiac assessment is only performed prior to and during the treatment period for HER-2 positive patients in Sweden, a longer-term cardiac monitoring program might be helpful for these patients.

Population-based estimates of incidence of ischemic heart disease following treatment with aromatase inhibitors have not been extensively reported in the literature (*Matthews et al., 2018*). Abdel-Qadir et al. also found an increased risk of myocardial infarction in patients treated with aromatase inhibitors, as compared to those treated with tamoxifen (*Abdel-Qadir et al., 2016*). It has been posited that this apparent increase in risk of ischemic heart disease might in fact be attributed to reduced risk of heart disease following tamoxifen treatment (*Hackshaw et al., 2011*; *Rosell et al., 2013*). Nonetheless, given that in the present study, an increased risk of ischemic heart disease was observed in patients treated with aromatase inhibitors compared to patients not receiving hormone treatment, the protective effect from tamoxifen could not fully explain the previously reported risk increase of ischemic heart disease in patients treated with aromatase inhibitors.

While several previous studies have shown that radiotherapy for breast cancer increase the risk of ischemic heart disease and heart failure (*Darby et al., 2013*; *Harris et al., 2006*; *Pinder et al., 2007*), some studies, particularly those including patients diagnosed in recent decades, have not seen such an increase (*Boekel et al., 2014*; *Wadsten et al., 2018*; *Wennstig et al., 2020*). Indeed, the time trends of heart dose improvement and the use of modern heart dose sparing techniques, together with individualized doses of therapy, may result in lower doses of radiation to the heart (*Taylor et al., 2017*; *Taylor and Kirby, 2015*; *Taylor et al., 2009*). Another reason for inconsistent results in the literature could be that patients with a left-sided breast cancer who are susceptible to cardiovascular diseases are less likely to be selected for radiotherapy (*Darby et al., 2013*; *Taylor and Kirby, 2015*). Therefore, in our study, we compared risk of ischemic heart disease in patients treated with radiotherapy for left-sided to those treated for right-sided breast cancer. We found slightly increased non-significant risk of ischemic heart disease, this risk appeared (as expected) more pronounced in the follow-up period after 10 years. Overall, our results indicate only small risk of heart disease due to radiotherapy in women treated in Sweden after year 2000. Further studies with detailed information on the mean heart dose of radiation or total cumulative radiation dose administered are therefore needed to confirm and provide more context to this finding.

A major strength of our study is that we were able to investigate risks of heart disease by time since diagnosis and adjuvant treatment. The risk estimates presented in this study therefore provide an insight into the short-term and long-term effects of adjuvant breast cancer therapy in routine clinical practice. Notably, all breast cancer patients were followed-up for a relatively long period. The

population-based nature of the study and linkage to registry-based data further minimized the risk of selection and information biases.

We acknowledge several limitations. The Swedish Patient Register has high validity for heart failure, arrhythmia, and ischemic heart disease (with positive predictive value between 88% and 98%) (*Hammar et al., 2001*; *Ludvigsson et al., 2011*), by analyzing main diagnoses only. However, misclassification of heart diseases may still have occurred. In addition, pre-existing comorbidities extracted from the patient register may not include those patients with mild symptoms. Second, we did not have complete data on trastuzumab during the study period and relied on HER-2 status as proxy, which may have resulted in some misclassification. The numbers of heart disease after treatment with trastuzumab and taxane were also relatively small in our study, and might have resulted in limited power or chance finding when identifying an association between trastuzumab/taxane and the relevant cardiac events. Besides, the Stockholm-Gotland Breast Cancer Register only records intended treatment, not whether patients actually received these therapies. However, the agreement between the intended and administered breast cancer treatment in Sweden has been previously reported to be about 95% (*Löfgren et al., 2019*). Finally, it should be noted that cancer patients are subject to increased medical surveillance, especially in the initial period after diagnosis. This could explain the increased rates of heart disease events especially within the first year of follow-up, although psychological distress following recent diagnosis with cancer represents an alternative plausible explanation for this observation (*Fang et al., 2012*; *Schoormans et al., 2016*).

Through this study, we demonstrate that compared to the general population, women with breast cancer have increased risks of heart disease including heart failure and arrhythmia. The short-term risk of ischemic heart disease diminished after 1 year post diagnosis. The increased risk of arrhythmia and heart failure however appears to persist long-term, beyond the first decade after diagnosis. Administration of systemic adjuvant therapies is associated with risks of heart disease. The risk estimates observed in this study may serve as reference to aid adjuvant therapy decision-making and patient counselling in oncology practices.

## Acknowledgements

Funding: This work was supported by the Swedish Research Council (grant no: 2018-02547); Swedish Cancer Society (grant no: CAN-19-0266); and FORTE (grant no: 2016-00081). HY was supported by Startup Fund for high-level talents of Fujian Medical University (grant no: XRCZX2020007) and Startup Fund for scientific research, Fujian Medical University (grant no: 2019QH1002). NBP was supported by the University of Malaya Impact-Oriented Interdisciplinary Research Grant Programme (grant no: IIRG006C-19HWB). EZ and WB was partly supported by China Scholarship council. Dr Ludvigsson coordinates a study on behalf of the Swedish IBD quality register (SWIBREG), which has received funding from Janssen corporation. The funders had no role in the study design, data collection, analyses, data interpretation, writing the manuscript, or in the decision to submit the manuscript for publication.

## Additional information

### Competing interests

Nirmala Bhoo-Pathy: received educational grants to their institution from Novartis, Pfizer, AIA Bhd and Pharmaceutical Association of Malaysia.Has received speaker's fees from Novartis, Pfizer and Roche, and received travel support from Roche and Pharmaceutical Association of Malaysia to attend conferences in 2018 and 2019. Has served on the advisory board of Pfizer Asia Pacific, Malaysia (2017/18 year), and been a committee member for Together Against Cancer (NGO) (2018 and 2019). Roche Diagnostics also provided Nirmala Bhoo-Pathy with research material, namely COVID-19 total antibody kits. The author has no other competing interests to declare. Jonas Bergh: research was supported by payments from Amgen, AstraZeneca, Bayer, Merck, Roche and Sanofi-Aventis to their institution, along with payments from non-profit organisations (Swedish Cancer Society and Knut Alice Wallenberg) and the Swedish Research Council. Also gave lectures to Astra Zeneca and Roche (no personal payment was received for these). Is a scientific advisor to The Medical product agency and

to EMA, and is a representative of Swedish Breast Cancer Group. The author has no other competing interests to declare. Jonas F Ludvigsson: coordinates a study on behalf of the Swedish IBD quality register (SWIBREG). This study has received funding from Janssen corporation. The author has no other competing interests to declare. The other authors declare that no competing interests exist.

## Funding

| Funder | Grant reference number | Author |
|---|---|---|
| Natural Science Foundation of Fujian Province | 2021J01721 | Haomin Yang |
| Startup Fund for High-level Talents of Fujian Medical University | XRCZX2020007 | Haomin Yang |
| Startup Fund for Scientific Research, Fujian Medical University | 2019QH1002 | Haomin Yang |
| Laboratory Construction Program of Fujian Medical University | 1100160208 | Haomin Yang |
| Vetenskapsrådet | 2018-02547 | Kamila Czene |
| Swedish Cancer Foundation | CAN-19-0266 | Kamila Czene |
| Forskningsrådet om Hälsa, Arbetsliv och Välfärd | 2016-00081 | Kamila Czene |
| University of Malaya | Impact-Oriented Interdisciplinary Research Grant Programme IIRG006C-19HWB | Nirmala Bhoo-Pathy |
| China Scholarship Council | | Erwei Zeng |
| China Scholarship Council | | Weiwei Bian |

The funders had no role in study design, data collection and interpretation, or the decision to submit the work for publication.

## Author contributions

Haomin Yang, Conceptualization, Data curation, Formal analysis, Funding acquisition, Investigation, Methodology, Project administration, Resources, Software, Validation, Visualization, Writing – original draft, Writing - review and editing; Nirmala Bhoo-Pathy, Conceptualization, Data curation, Formal analysis, Funding acquisition, Methodology, Project administration, Writing – original draft; Judith S Brand, Conceptualization, Data curation, Formal analysis, Investigation, Resources, Software, Writing – original draft; Elham Hedayati, Conceptualization, Methodology, Resources, Writing – original draft; Felix Grassmann, Formal analysis, Software, Writing – original draft; Erwei Zeng, Formal analysis, Validation, Writing – original draft; Jonas Bergh, Per Hall, Conceptualization, Data curation, Supervision, Writing – original draft; Weiwei Bian, Data curation, Formal analysis, Validation, Writing – original draft; Jonas F Ludvigsson, Conceptualization, Supervision, Validation, Writing – original draft; Kamila Czene, Conceptualization, Data curation, Funding acquisition, Resources, Supervision, Writing – original draft, Writing - review and editing

## Author ORCIDs

Haomin Yang http://orcid.org/0000-0002-2252-2606
Nirmala Bhoo-Pathy http://orcid.org/0000-0003-0568-8863
Felix Grassmann http://orcid.org/0000-0003-1390-7528
Per Hall http://orcid.org/0000-0002-5640-9126

## Ethics

Human subjects: The study was approved by the Regional Ethical Review Board in Stockholm (Dnr 2009/254-31/4). In accordance with their decision, it was not necessary to obtain informed consent

from participants involved in the study. All individuals' information was anonymized and de-identified prior to analysis.

### Decision letter and Author response
Decision letter https://doi.org/10.7554/eLife.71562.sa1
Author response https://doi.org/10.7554/eLife.71562.sa2

## Additional files

### Supplementary files
- Transparent reporting form
- Source code 1. STATA script for the analyses.

### Data availability
The data used in this study are owned by the Swedish National Board of Health and Welfare and Statistics Sweden. According to Swedish law and GDPR, the authors are not able to make the dataset publicly available. Any researchers (including international researchers) interested in obtaining the data can do so by the following steps: (1) apply for ethical approval from their local ethical review boards; (2) contact the Swedish National Board of Health and Welfare and/or Statistics Sweden with the ethical approval and make a formal application of use of register data.

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

## Appendix 1

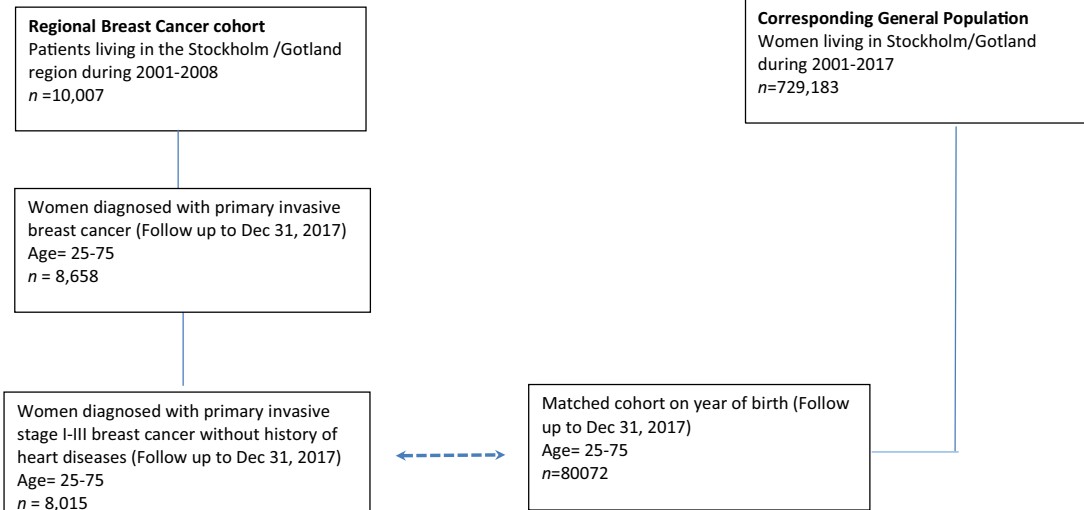

**Appendix 1—figure 1.** Flowchart of the data.

## Appendix 2

**Appendix 2—table 1.** ICD codes used in the analyses.

| Disease codes | ICD-10 | ICD-9 |
|---|---|---|
| Arrhythmia | I47, I48, I49 | 427 |
| Atrial fibrillation | I48 | 427D |
| Tachycardia and other cardiac arrhythmias | I47, I49 | 427A–C, 427E–J |
| Heart failure | I50 | 428A, 428B, 428X |
| Ischemic heart disease | I20–I25 | 410–414 |
| Myocardial infarction | I21, I22 | 410 |
| Angina pectoris | I20 | 411B, 413, 414A |
| Hypertension | I10–I15 | 401–405 |
| Tobacco abuse and Chronic Pulmonary Disease | J40–J47, J60–J67, F17, I278, I279, J684, J701, J703 | 490–496,500–505, 416W, 416X, 506E, 508B, 508W, 305B |

ICD = International Classification of Diseases.

**Appendix 2—table 2.** Cumulative incidence estimates of heart diseases in breast cancer patients and age matched controls.

| | Cumulative incidence, % (95% CI) | | | | | |
|---|---|---|---|---|---|---|
| | **6 months** | **1 year** | **2 years** | **5 years** | **10 years** | **15 years** |
| Arrhythmia | | | | | | |
| Breast cancer | 0.49 (0.35–0.66) | 0.81 (0.63–1.02) | 1.25 (1.02–1.52) | 2.77 (2.41–3.16) | 6.04 (5.49–6.63) | 11.03 (10.04–12.08) |
| Matched control | 0.20 (0.17–0.23) | 0.38 (0.34–0.42) | 0.79 (0.73–0.85) | 2.18 (2.08–2.29) | 4.98 (4.83–5.13) | 8.21 (7.99–8.45) |
| Heart failure | | | | | | |
| Breast cancer | 0.09 (0.04–0.18) | 0.28 (0.18–0.42) | 0.53 (0.38–0.71) | 1.06 (0.85–1.32) | 2.32 (1.98–2.70) | 4.80 (4.13–5.54) |
| Matched control | 0.05 (0.03–0.06) | 0.10 (0.08–0.13) | 0.22 (0.19–0.26) | 0.68 (0.63–0.74) | 1.87 (1.77–1.96) | 3.78 (3.62–3.95) |
| Ischemic heart disease | | | | | | |
| Breast cancer | 0.26 (0.17–0.40) | 0.48 (0.35–0.65) | 0.92 (0.73–1.16) | 1.93 (1.64–2.27) | 3.59 (3.17–4.05) | 5.68 (4.98–6.44) |
| Matched control | 0.13 (0.11–0.16) | 0.33 (0.29–0.37) | 0.72 (0.67–0.78) | 1.91 (1.81–2.00) | 3.85 (3.72–3.99) | 5.90 (5.71–6.09) |

Abbreviations: CI = confidence interval. Cumulative incidence estimates of heart diseases in all breast cancer patients (N = 8015) at different time-points following the index date ( = date of diagnosis in breast cancer patients). Cumulative incidence estimates are obtained from Aalen-Johansen estimation.

**Appendix 2—table 3.** Hazard ratios for heart diseases by different adjuvant therapies and time since diagnosis.

| | HR (95% CI) for arrhythmia | | HR (95% CI) for heart failure | | HR (95% CI) for ischemic heart disease | |
|---|---|---|---|---|---|---|
| Treatment variables | <10 years | >10 years | <10 years | >10 years | <10 years | >10 years |
| Radiotherapy | | | | | | |
| Right-sided | REF (1.00) | REF (1.00) | REF (1.00) | REF (1.00) | REF (1.00) | REF (1.00) |
| Left-sided | 0.99 (0.79–1.24) | 0.80 (0.57–1.14) | 1.13 (0.77–1.66) | 0.94 (0.56–1.56) | 1.13 (0.84–1.53) | 1.29 (0.70–2.37) |
| Both-sided | 0.26 (0.04–1.90) | 1.01 (0.30–3.36) | 1.49 (0.36–6.24) | 2.37 (0.54–10.41) | 0.84 (0.20–3.43) | 1.64 (0.21–12.89) |
| Chemotherapy | | | | | | |

*Appendix 2—table 3 Continued on next page*

*Appendix 2—table 3 Continued*

| | HR (95% CI) for arrhythmia | | HR (95% CI) for heart failure | | HR (95% CI) for ischemic heart disease | |
|---|---|---|---|---|---|---|
| No | REF (1.00) | REF (1.00) | REF (1.00) | REF (1.00) | REF (1.00) | REF (1.00) |
| Anthracyclines-based | 1.08 (0.81–1.46) | 1.07 (0.65–1.76) | **1.75 (1.11–2.76)** | 1.66 (0.86–3.19) | 1.27 (0.87–1.85) | 1.36 (0.60–3.09) |
| Anthracyclines + taxanes | 1.02 (0.51–2.01) | 0.87 (0.13–5.77) | **3.72 (1.72–8.03)** | 0.00 (0.00–0.00) | 1.38 (0.60–3.18) | 0.00 (0.00–0.00) |
| CMF | 1.24 (0.59–2.61) | 0.74 (0.18–3.15) | 1.16 (0.33–4.05) | 0.00 (0.00–0.00) | 0.97 (0.33–2.84) | 1.17 (0.16–8.41) |
| Hormone therapy | | | | | | |
| No | REF (1.00) | REF (1.00) | REF (1.00) | REF (1.00) | REF (1.00) | REF (1.00) |
| Tamoxifen | 0.91 (0.67–1.23) | 1.08 (0.67–1.73) | 0.76 (0.49–1.20) | 0.98 (0.52–1.84) | 0.93 (0.62–1.39) | 1.50 (0.64–3.53) |
| Aromatase inhibitors | 0.97 (0.70–1.36) | 1.05 (0.56–1.95) | 0.93 (0.58–1.49) | 1.47 (0.66–3.27) | 1.47 (0.96–2.24) | 1.31 (0.43–4.01) |
| Trastuzumab * | | | | | | |
| No | REF (1.00) | REF (1.00) | REF (1.00) | REF (1.00) | REF (1.00) | REF (1.00) |
| Yes | 1.56 (0.82–2.97) | 0.00 (0.00–0.00) | **2.60 (1.14–5.89)** | 0.00 (0.00–0.00) | 1.71 (0.76–3.87) | 6.95 (0.56–85.58) |

HR = hazard ratio; CI = confidence interval. Hazard ratios are estimated from Cox proportional hazards models with time since diagnosis as underlying time scale. Hazard ratios are multivariable adjusted including age at diagnosis, year of diagnosis, menopausal status, Charlson comorbidity index, clinical stage, type of surgery, hypertension, chronic pulmonary disease and tobacco abuse, and all treatment variables listed in the table. * Treatment-specific analysis of trastuzumab was restricted to patients diagnosed between 2005 and 2008.

## Appendix 3

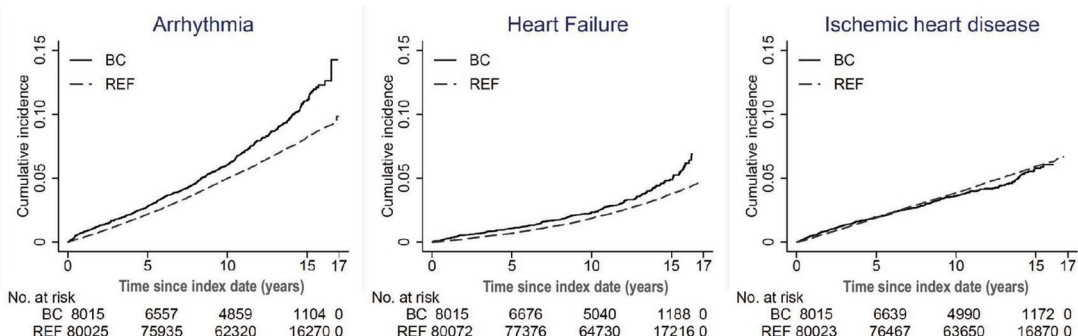

**Appendix 3—figure 1.** Cumulative incidence of heart disease in breast cancer patients and matched women. Aalen-Johansen estimates of the cumulative risk of heart disease by time since index date, in breast cancer patients and matched women from the general population. BC: breast cancer; REF: reference women in the matched controls from the general population; No.: number.

