## [Editor Report]

We feel that your work will be of interest to breast medical oncologists, cardiologists, and primary care providers who treat patients with breast cancer. We commend you for this study, which achieves its goal of identifying the incidence and hazard ratio of cardio-toxicity associated with breast cancer treatment within a general breast cancer population. The international nature of your collaborative study along with its large patient cohort size and long horizontal follow up are quite attractive features in solidifying previous findings and discovering future areas of exploration.

---

## [Decision Letter]

**Decision letter after peer review:**

Thank you very much for submitting your work to *eLife*. Your article has been reviewed by 3 peer reviewers and the evaluation has been overseen by Eduardo Franco as the Senior Editor. The following individual involved in review of your submission has agreed to reveal their identity: Philip Boonstra (Reviewer #3).

As you will see in the attached reviews, while the Referees generally recognized the merits of your draft, several technical points of concern in your methodology and findings were also highlighted. Thus, we remain interested potentially publishing an extensively revised version of your manuscript. However, we would like you to specifically address the following perceived weaknesses of your work, in particular the biostatistical methodology, as follows:

Essential revisions:

1. The title of the draft suggests the risk of heart disease is associated with breast cancer itself, but the content of the manuscript and conclusions emphasize the risk based upon treatment effects. Please, modify the title to reflect the risk of breast cancer therapy.

2. For Table 1: In addition to Charlson Comorbidity Index (CCI score), please include CVD risk factors (e.g., HTN, HLD, DM2, BMI, tobacco-smoking) to help readers understand the baseline cardiovascular risk of your patients. This criticism assumes importance as these risk factors were used in the risk models, and independent of cancer treatment are known to be associated with risk of arrhythmias, ischemic heart disease, and congestive heart failure.

3. For Table 2: Please add the word "Ratio" i.e. "Hazard ratio for heart diseases" in the title.

4. The finding of no significant increased risk of ischemic heart disease after left breast radiation is quite interesting. This provocative result would become more powerful (and better supported) if estimates of mean heart dose of radiation or even total cumulative radiation dose administered were included. If those may not be available from the registry data, please discuss this potentially counterintuitive finding.

5. Regarding hypotheses made in the Discussion section: The authors appropriately provide citations regarding early increased risk of ischemic heart disease due to emotional distress, however there are several other factors that could potentially increase this risk that warrant consideration, namely: surgery for breast cancer (98.9% of patients in this cohort) typically takes place within the first year of diagnosis and may increase risk of arrhythmia, ischemic event; patients with cancer are at increased risk of arterial thromboembolism (ATE) which includes myocardial infarction 150 days prior to cancer diagnosis and this risk appears to attenuate 1 year after diagnosis. (Navi, B. B., et al., (2017). "Risk of Arterial Thromboembolism in Patients with Cancer." J Am Coll Cardiol 70(8): 926-938) (Navi, B. B., et al., (2019). "Arterial thromboembolic events preceding the diagnosis of cancer in older persons." Blood 133(8): 781-789.)

6. Please discuss the fact that while the results are generally interesting and hypothesis-generating, the patient population is overall young and healthy (median age 59, majority CCI = 0); thus, one should be cautious to extrapolate results to guide individual therapy decisions in clinical practice.

7. It is also unclear whether there was any protocol in place for cardiac monitoring for patients receiving cardiotoxic chemotherapy or Anti Her2neu agents. Please clarify it in the revised draft, either way.

8. With regard to the matched analysis of time to heart disease diagnosis: For the breast cancer cohort, were patients with a diagnosis of heart disease prior to cancer diagnosis included in the analysis? If so, how was the event (which precedes time = 0) incorporated into the analysis? If not, please make sure to make note of this important restriction. Please, keep in mind that Referee 3 clearly favors the latter approach.

9. Moreover, for the matched cohort: What is time = 0 for these persons? i.e. how does one interpret "Time since diagnosis" on Figure 1 for a patient who has not been diagnosed with breast cancer?

10. Finally, for the matched cohort: How was the matching incorporated into the FPM? Presumably there should be a frailty term of some sort to indicate the matched groups, within which there is expected to be correlation.

11. Kaplan-Meier curves were used to estimate the cumulative incidence of heart disease. How was death of the patient prior to diagnosis of heart disease handled? Of note, Referee #3 argues that Kaplan-Meier is not the best analytical approach here because Kaplan-Meier tends to overestimate the event rate when competing events are counted as censoring. In this setting, Referee #3 favors an Aaalen-Johansen-type estimator, which treats death as a competing event. For instance, please see: https://pubmed.ncbi.nlm.nih.gov/10204198/

12. Please address and correct: The sentence "Missing indicators were included for the analysis of these covariates in the model" and the results in Table 3 suggest that some missing values were analyzed "as is", meaning that "missingness" was used as a category itself. This, of course, is not desirable and there exists methodology+software for more appropriately handling these data, e.g. multiple imputation with chained equations. For example, how does one interpret that "unknown chemotherapy" status is positively associated with heart failure but less so than anthracycline-based chemotherapy.

13. Please address and correct: The reported HRs (at the top of p. 10) seem incongruous with the FPM model demonstrated in Figure 1, since there is clearly a non-linear relationship between the hazard and the outcome.

14. Please address and correct: It seems unlikely that breast cancer diagnosis could ever be "protective" for ischemic heart disease. A more constrained model that does not allow for the possibility of HR < 1 could provide a more sensible estimate of this time-dependent HR.

15. Please address and correct: As an alternative to a four-category radiotherapy variable, which (as the authors note) requires assuming that bilateral radiotherapy is equivalent to left-sided radiotherapy, it would seem sensible to create two separate binary variables (left, y vs. no and right, y vs. no).

16. Please double-check rows two and three of the first column of Table 2. One would expect the HRs for disease history (No, 1.28 and Yes, 1.30) to fall on either side of the overall HR (1.27), but they don't: Is this Simpson's paradox or a mistake or something else? Please verify and clarify.

*Reviewer #1 (Recommendations for the authors):*

Thank you for the opportunity to review this manuscript.

– The title of the paper suggests the risk of heart disease is associated with breast cancer itself, but the content of the manuscript and conclusions emphasize the risk based upon treatment effects. I would consider changing the title to reflect the risk of breast cancer therapy.

– For Table 1: In addition to Charlson Comorbidity Index (CCI score), would also include CVD risk factors (HTN, HLD, DM2, BMI, tobacco smoking) to better understand patients' baseline cardiovascular risk, particularly as these risk factors were used in the risk models, and independent of cancer treatment are known to be associated with risk of arrhythmias, ischemic heart disease, and congestive heart failure.

– For Table 2: Would add the word "Ratio" ie "Hazard ratio for heart diseases" in the title.

– The finding of no significant increased risk of ischemic heart disease after left breast radiation is quite interesting. This finding would be more powerful and better supported if estimates of mean heart dose of radiation or even total cumulative radiation dose administered was included (which may not be possible from the available registry data).

– Regarding hypotheses made in the Discussion section: the authors appropriately provide citations regarding early increased risk of ischemic heart disease due to emotional distress, however there are several other factors that could potentially increase this risk that warrant consideration: surgery for breast cancer (98.9% of patients in this cohort) typically takes place within the first year of diagnosis and may increase risk of arrhythmia, ischemic event; patients with cancer are at increased risk of arterial thromboembolism (ATE) which includes myocardial infarction 150 days prior to cancer diagnosis and this risk appears to attenuate 1 year after diagnosis. (Navi, B. B., et al., (2017). "Risk of Arterial Thromboembolism in Patients With Cancer." J Am Coll Cardiol 70(8): 926-938) (Navi, B. B., et al., (2019). "Arterial thromboembolic events preceding the diagnosis of cancer in older persons." Blood 133(8): 781-789.)

– While the results are interesting and hypothesis generating, the patient population is overall young and healthy (median age 59, majority CCI = 0) therefore would be cautious to recommend extrapolating results to guide individual therapy decisions in clinical practice.

*Reviewer #3 (Recommendations for the authors):*

As an alternative to a four-category radiotherapy variable, which as the authors note requires assuming that bilateral radiotherapy is equivalent to leftsided radiotherapy, it would seem sensible to create two separate binary variables (left, y vs. no and right, y vs. no).

Can the authors double check rows two and three of the first column of Table 2? Intuitively, I would expect the HRs for disease history (No, 1.28 and Yes, 1.30) to fall on either side of the overall HR (1.27), but they don't. Is this Simpsons paradox or a mistake or something else?

---

## [Author Response]

Essential revisions:1. The title of the draft suggests the risk of heart disease is associated with breast cancer itself, but the content of the manuscript and conclusions emphasize the risk based upon treatment effects. Please, modify the title to reflect the risk of breast cancer therapy.

As suggested by the reviewer, we have revised the title of the paper to “Risk of heart disease following treatment for breast cancer: results from a population-based cohort study”.

2. For Table 1: In addition to Charlson Comorbidity Index (CCI score), please include CVD risk factors (e.g., HTN, HLD, DM2, BMI, tobacco-smoking) to help readers understand the baseline cardiovascular risk of your patients. This criticism assumes importance as these risk factors were used in the risk models, and independent of cancer treatment are known to be associated with risk of arrhythmias, ischemic heart disease, and congestive heart failure.

In Table 1, we have now added the numbers for the risk factors used in the risk models (history of hypertension, chronic pulmonary disease and tobacco abuse). Diabetes was already included in the Charlson Comorbidity Index, so we did not list diabetes separately.

3. For Table 2: Please add the word "Ratio" i.e. "Hazard ratio for heart diseases" in the title.

We have revised the title for Table 2 to “Hazard ratios for heart diseases in breast cancer patients compared to the matched cohort.”

4. The finding of no significant increased risk of ischemic heart disease after left breast radiation is quite interesting. This provocative result would become more powerful (and better supported) if estimates of mean heart dose of radiation or even total cumulative radiation dose administered were included. If those may not be available from the registry data, please discuss this potentially counterintuitive finding.

Unfortunately, information on the mean heart dose of radiation or total cumulative radiation dose administered was not recorded in the Stockholm-Gotland Breast Cancer Register. We therefore cannot analyze this. We have now discussed this finding in more detail in the Discussion section.

In discussion, page 14-15, line 308-320

“Indeed, the time trends of heart dose improvement and the use of modern heart dose sparing techniques, together with individualized doses of therapy, may result in lower doses of radiation to the heart (C. Taylor et al., 2017; C. W. Taylor and Kirby, 2015; C. W. Taylor et al., 2009). Another reason for inconsistent results in the literature could be that patients with a left sided breast cancer who are susceptible to cardiovascular disease are less likely to be selected for radiotherapy (C. W. Taylor and Kirby, 2015). Therefore, in our study, we compared risk of ischemic heart disease in patients treated with radiotherapy for left-sided to those treated for right-sided breast cancer. We found slightly increased non-significant risk of ischemic heart disease, this risk appeared (as expected) more pronounced in the follow-up period after 10 years. Overall, our results indicate only small risk of heart disease due to radiotherapy in women treated in Sweden after year 2000. Further studies with detailed information on the mean heart dose of radiation or total cumulative radiation dose administered are therefore needed to confirm and provide more context to this finding.”

5. Regarding hypotheses made in the Discussion section: The authors appropriately provide citations regarding early increased risk of ischemic heart disease due to emotional distress, however there are several other factors that could potentially increase this risk that warrant consideration, namely: surgery for breast cancer (98.9% of patients in this cohort) typically takes place within the first year of diagnosis and may increase risk of arrhythmia, ischemic event; patients with cancer are at increased risk of arterial thromboembolism (ATE) which includes myocardial infarction 150 days prior to cancer diagnosis and this risk appears to attenuate 1 year after diagnosis. (Navi, B. B., et al., (2017). "Risk of Arterial Thromboembolism in Patients with Cancer." J Am Coll Cardiol 70(8): 926-938) (Navi, B. B., et al., (2019). "Arterial thromboembolic events preceding the diagnosis of cancer in older persons." Blood 133(8): 781-789.)

We thank the reviewer for providing this potential alternative explanation and have now added this in our Discussion.

In discussion, page 12, line 264-271

“The finding that risk of ischemic heart disease in breast cancer patients was only transiently elevated after diagnosis is not unexpected, considering the emotional distress of dealing with a new cancer diagnosis in the patients, which may lead to higher short-term rates of ischemic heart disease (Fang et al., 2012; Schoormans, Pedersen, Dalton, Rottmann, and van de Poll-Franse, 2016). In addition, surgery after breast cancer diagnosis might increase the risk of arterial thromboembolism (Gervaso, Dave, and Khorana, 2021), which includes myocardial infarction, and the effect appears to attenuate one year after diagnosis. (Navi et al., 2017; Navi et al., 2019)”

6. Please discuss the fact that while the results are generally interesting and hypothesis-generating, the patient population is overall young and healthy (median age 59, majority CCI = 0); thus, one should be cautious to extrapolate results to guide individual therapy decisions in clinical practice.

Patients involved in this analysis were obtained from the Stockholm-Gotland Breast Cancer Register, covering more than 99% of the breast cancer patients in the Stockholm-Gotland region. As the reviewer noticed, the median age at diagnosis was 59, and the majority of them had no comorbidities. Therefore, caution is needed to generalize the findings in this study to very old patients with multiple comorbidities. We have now added this in the discussion.

In discussion, page 12, line 246-251

“We found an increased risk of arrhythmia and heart failure in breast cancer patients as compared with the matched reference individuals from the general population, which is similar to the risk of heart failure reported by a previous Dutch study (Hooning et al., 2007), indicating the generalizability of our findings to European countries. However, as patients in our cohort were aged between 25 and 75 years, caution is needed when generalizing these findings to older patients, who may have more comorbidities.”

7. It is also unclear whether there was any protocol in place for cardiac monitoring for patients receiving cardiotoxic chemotherapy or Anti Her2neu agents. Please clarify it in the revised draft, either way.

In Sweden, there is no cardiac monitoring for chemotherapy in routine clinical practice. For HER2-therapy, cardiac monitoring with a thorough cardiac assessment prior to treatment, including history, physical examination, and determination of left ventricular ejection fraction before, during and right after treatment has been mandatory since introduction in clinical routine. We have now added this information to the discussion.

In discussion, page 13, line 288-291

“As there is no cardiac monitoring for chemotherapy in routine clinical practice and cardiac assessment is only performed prior to and during the treatment period for HER-2 positive patients in Sweden, a longer-term cardiac monitoring program might be helpful for these patients.”

8. With regard to the matched analysis of time to heart disease diagnosis: For the breast cancer cohort, were patients with a diagnosis of heart disease prior to cancer diagnosis included in the analysis? If so, how was the event (which precedes time = 0) incorporated into the analysis? If not, please make sure to make note of this important restriction. Please, keep in mind that Referee 3 clearly favors the latter approach.

As suggested by Referee 3, we have now excluded those patients with a diagnosis of heart disease prior to cancer diagnosis. We have updated the results and the methods section accordingly.

In Materials and methods, page 6, line 106-108

“We included all patients diagnosed with non-metastatic breast cancer (stages I-III) and without prior diagnosis of heart disease at age 25 to 75 years (N = 8015).”

9. Moreover, for the matched cohort: What is time = 0 for these persons? i.e. how does one interpret "Time since diagnosis" on Figure 1 for a patient who has not been diagnosed with breast cancer?

We apologize for this misunderstanding and have revised it to “Time since index date ( = date of diagnosis, which is the same date for corresponding matched individual from the general population) ” in Figure 1.

10. Finally, for the matched cohort: How was the matching incorporated into the FPM? Presumably there should be a frailty term of some sort to indicate the matched groups, within which there is expected to be correlation.

In the flexible parametric survival model for matched cohort data, a shared frailty term was incorporated into the model to indicate the matched cluster. The maximum (penalized) marginal likelihood method is used to estimate the regression coefficients and the variance for the frailty. We have added this explanation in the methods part.

In Materials and methods, page 9, line 167-170

“Considering the correlation within the matched clusters, a shared frailty term (as random effects) was incorporated into the model and the maximum (penalized) marginal likelihood method was used to estimate the regression coefficients and the variance for the frailty.”

11. Kaplan-Meier curves were used to estimate the cumulative incidence of heart disease. How was death of the patient prior to diagnosis of heart disease handled? Of note, Referee #3 argues that Kaplan-Meier is not the best analytical approach here because Kaplan-Meier tends to overestimate the event rate when competing events are counted as censoring. In this setting, Referee #3 favors an Aaalen-Johansen-type estimator, which treats death as a competing event. For instance, please see: https://pubmed.ncbi.nlm.nih.gov/10204198/

As suggested by the reviewer, we have now used the Aalen-Johansen method to estimate the cumulative incidence of heart disease and revised the text in the Methods, as well as the tables and figures in the supplement.

In Materials and methods, page 9, line 170-172

“Aalen-Johansen estimation was used to assess the cumulative incidences of heart diseases in breast cancer patients and matched reference individuals, while other causes of death were considered as competing events.”

12. Please address and correct: The sentence "Missing indicators were included for the analysis of these covariates in the model" and the results in Table 3 suggest that some missing values were analyzed "as is", meaning that "missingness" was used as a category itself. This, of course, is not desirable and there exists methodology+software for more appropriately handling these data, e.g. multiple imputation with chained equations. For example, how does one interpret that "unknown chemotherapy" status is positively associated with heart failure but less so than anthracycline-based chemotherapy.

Missingness of the type of adjuvant treatment was considered as a category in the previous version of our manuscript. To address potential biases resulting from missing data, we have now used multiple imputation with chained equations and revised the methods and Table 3 accordingly.

In Materials and methods, page 9, line 183-185

“Multiple imputation with chained equations was used to deal with the treatment categories with missing information. We replaced the missing data with 10 rounds of imputations and all the covariates were included in the imputation model.”

13. Please address and correct: The reported HRs (at the top of p. 10) seem incongruous with the FPM model demonstrated in Figure 1, since there is clearly a non-linear relationship between the hazard and the outcome.

As shown in the FPM model in Figure 1, HRs were not constant according to time since index date. Therefore, in the revised version, we only showed the HRs separately in <1, 1-2, 2-5, 5-10 and 10-17 years after diagnosis. We have revised the abstract, methods, and Table 2.

In Abstract, page 3, line 52-58:

“Time-dependent analyses revealed long-term increased risks of arrhythmia and heart failure following breast cancer diagnosis. Hazard ratios (HRs) within the first year of diagnosis were 2.14 (95% CI = 1.63-2.81) for arrhythmia and 2.71 (95% CI = 1.70-4.33) for heart failure. HR more than 10 years following diagnosis was 1.42 (95% CI = 1.21-1.67) for arrhythmia and 1.28 (95% CI = 1.03-1.59) for heart failure. The risk for ischemic heart disease was significantly increased only during the first year after diagnosis (HR=1.45, 95% CI = 1.03-2.04).”

In Materials and methods, page 8, line 159-161:

“We compared the risk of heart diseases in breast cancer patients with that observed in the matched cohort, using flexible parametric model (FPM) with time since index date as underlying time scale.”

In Results, page 10, line 213-216:

“A short-term increase in risks of arrhythmia and heart failure was found in breast cancer patients (Table 2, Figure 1, HR at first year for arrhythmia = 2.14; 95% CI = 1.63-2.81, for heart failure = 2.71; 95% CI = 1.70-4.33, respectively).”

14. Please address and correct: It seems unlikely that breast cancer diagnosis could ever be "protective" for ischemic heart disease. A more constrained model that does not allow for the possibility of HR < 1 could provide a more sensible estimate of this time-dependent HR.

To the best of our knowledge, the inverse association between breast cancer and the long-term risk of ischemic heart disease is possible considering that some of the reproductive risk factors for breast cancer have protective effect on the risk of ischemic heart disease. We have now discussed about this in Discussion.

In Discussion, page 13, line 271-276

“The long term lower risk of ischemic heart disease in breast cancer patients compared to age-matched women might be explained by the opposite role of reproductive factors in breast cancer and ischemic heart disease. Women with younger age at menarche and older age at menopause were associated with increased risk of breast cancer, while decreased risk of ischemic heart disease were found among these women (Collaborative Group on Hormonal Factors in Breast, 2012; Okoth et al., 2020).”

15. Please address and correct: As an alternative to a four-category radiotherapy variable, which (as the authors note) requires assuming that bilateral radiotherapy is equivalent to left-sided radiotherapy, it would seem sensible to create two separate binary variables (left, y vs. no and right, y vs. no).

As we have explained in the Methods section, there could be selection bias in the administration of radiotherapy to the patients. Therefore, the analysis for radiotherapy only included patients receiving radiotherapy. To show the effect of bilateral radiotherapy, we categorized these patients into a separate group, and revised the Methods section and Table 3.

In Materials and methods, page 7, line 139-141

“Since radiotherapy to the left breast has, in particular, been implicated in heart complications, radiotherapy was categorized according to tumor laterality (left vs. right). Bilateral tumors were coded separately in this analysis.”

In Materials and methods, page 9, line 180-183

“Considering the possible selection bias in the administration of radiotherapy (Wadsten et al., 2018), the analysis for radiotherapy only included patients receiving radiotherapy, making a comparison between left-sided, right-sided and both-sided breast cancer.”

16. Please double-check rows two and three of the first column of Table 2. One would expect the HRs for disease history (No, 1.28 and Yes, 1.30) to fall on either side of the overall HR (1.27), but they don't: Is this Simpson's paradox or a mistake or something else? Please verify and clarify.

There was a typing error in the previous version. In the revised version of manuscript, we do not present the overall HRs as suggested by the reviewers and editor.

Reviewer #1 (Recommendations for the authors):Thank you for the opportunity to review this manuscript.– The title of the paper suggests the risk of heart disease is associated with breast cancer itself, but the content of the manuscript and conclusions emphasize the risk based upon treatment effects. I would consider changing the title to reflect the risk of breast cancer therapy.

As suggested by the reviewer, we have revised the title of the paper to “Risk of heart disease following treatment for breast cancer: results from a population-based cohort study”.

– For Table 1: In addition to Charlson Comorbidity Index (CCI score), would also include CVD risk factors (HTN, HLD, DM2, BMI, tobacco smoking) to better understand patients' baseline cardiovascular risk, particularly as these risk factors were used in the risk models, and independent of cancer treatment are known to be associated with risk of arrhythmias, ischemic heart disease, and congestive heart failure.

In Table 1, we have now added the numbers for the risk factors used in the risk models (history of hypertension, chronic pulmonary disease and tobacco abuse). Diabetes was already included in the Charlson Comorbidity Index, so we did not list diabetes separately.

– For Table 2: Would add the word "Ratio" ie "Hazard ratio for heart diseases" in the title.

We have revised the title for Table 2 to “Hazard ratios for heart diseases in breast cancer patients compared to the matched cohort.”

– The finding of no significant increased risk of ischemic heart disease after left breast radiation is quite interesting. This finding would be more powerful and better supported if estimates of mean heart dose of radiation or even total cumulative radiation dose administered was included (which may not be possible from the available registry data).

Unfortunately, information on the mean heart dose of radiation or total cumulative radiation dose administered was not recorded in the Stockholm-Gotland Breast Cancer Register. We therefore cannot analyze this. We have now discussed this finding in more detail in the discussion section.

In discussion, page 14-15, line 308-320

“Indeed, the time trends of heart dose improvement and the use of modern heart dose sparing techniques, together with individualized doses of therapy, may result in lower doses of radiation to the heart (C. Taylor et al., 2017; C. W. Taylor & Kirby, 2015; C. W. Taylor et al., 2009). Another reason for inconsistent results in the literature could be that patients with a left sided breast cancer who are susceptible to cardiovascular disease are less likely to be selected for radiotherapy (C. W. Taylor & Kirby, 2015). Therefore, in our study, we compared risk of ischemic heart disease in patients treated with radiotherapy for left-sided to those treated for right-sided breast cancer. We found slightly increased non-significant risk of ischemic heart disease, this risk appeared (as expected) more pronounced in the follow-up period after 10 years. Overall, our results indicate only small risk of heart disease due to radiotherapy in women treated in Sweden after year 2000. Further studies with detailed information on the mean heart dose of radiation or total cumulative radiation dose administered are therefore needed to confirm and provide more context to this finding. ”

– Regarding hypotheses made in the Discussion section: the authors appropriately provide citations regarding early increased risk of ischemic heart disease due to emotional distress, however there are several other factors that could potentially increase this risk that warrant consideration: surgery for breast cancer (98.9% of patients in this cohort) typically takes place within the first year of diagnosis and may increase risk of arrhythmia, ischemic event; patients with cancer are at increased risk of arterial thromboembolism (ATE) which includes myocardial infarction 150 days prior to cancer diagnosis and this risk appears to attenuate 1 year after diagnosis. (Navi, B. B., et al., (2017). "Risk of Arterial Thromboembolism in Patients With Cancer." J Am Coll Cardiol 70(8): 926-938) (Navi, B. B., et al., (2019). "Arterial thromboembolic events preceding the diagnosis of cancer in older persons." Blood 133(8): 781-789.)

We thank the reviewer for providing this potential alternative explanation and have now added this in our Discussion.

In discussion, page 12, line 264-271

“The finding that risk of ischemic heart disease in breast cancer patients was only transiently elevated after diagnosis is not unexpected, considering the emotional distress of dealing with a new cancer diagnosis in the patients, which may lead to higher short-term rates of ischemic heart disease (Fang et al., 2012; Schoormans, Pedersen, Dalton, Rottmann, & van de Poll-Franse, 2016). In addition, surgery after breast cancer diagnosis might increase the risk of arterial thromboembolism (Gervaso, Dave, & Khorana, 2021), which includes myocardial infarction, and the effect appears to attenuate one year after diagnosis. (Navi et al., 2017; Navi et al., 2019)”

– While the results are interesting and hypothesis generating, the patient population is overall young and healthy (median age 59, majority CCI = 0) therefore would be cautious to recommend extrapolating results to guide individual therapy decisions in clinical practice.

Patients involved in this analysis were obtained from the Stockholm-Gotland Breast Cancer Register, covering more than 99% of the breast cancer patients in the Stockholm-Gotland region. As the reviewer noticed, the median age at diagnosis was 59, and the majority of them had no comorbidities. Therefore, caution is needed to generalize the findings in this study to very old patients with multiple comorbidities. We have now added this in the discussion.

In discussion, page 12, line 246-251

“We found an increased risk of arrhythmia and heart failure in breast cancer patients as compared with the matched reference individuals from the general population, which is similar to the risk of heart failure reported by a previous Dutch study (Hooning et al., 2007), indicating the generalizability of our findings to European countries. . However, as patients in our cohort were aged between 25 and 75 years, caution is needed when generalizing these findings to older patients, who may have more comorbidities.”

Reviewer #3 (Recommendations for the authors):As an alternative to a four-category radiotherapy variable, which as the authors note requires assuming that bilateral radiotherapy is equivalent to leftsided radiotherapy, it would seem sensible to create two separate binary variables (left, y vs. no and right, y vs. no).

As we have explained in the Methods section, there could be selection bias in the administration of radiotherapy to the patients. Therefore, the analysis for radiotherapy only included patients receiving radiotherapy. To show the effect of bilateral radiotherapy, we categorized these patients into a separate group, and revised the Methods section and Table 3.

In Materials and Methods, page 7, line 139-141

“Since radiotherapy to the left breast has, in particular, been implicated in heart complications, radiotherapy was categorized according to tumor laterality (left vs. right). Bilateral tumors were coded separately in this analysis.”

In Materials and Methods, page 9, line 180-183

“Considering the possible selection bias in the administration of radiotherapy (Wadsten et al., 2018), the analysis for radiotherapy only included patients receiving radiotherapy, making a comparison between left-sided, right-sided and both-sided breast cancer.”

Can the authors double check rows two and three of the first column of Table 2? Intuitively, I would expect the HRs for disease history (No, 1.28 and Yes, 1.30) to fall on either side of the overall HR (1.27), but they don't. Is this Simpsons paradox or a mistake or something else?

There was a typing error in the previous version. In the revised version of manuscript, we do not present the overall HRs as suggested by the reviewers and editor.